# 3′dNTP Binding Is Modulated during Primer Synthesis and Translesion by Human PrimPol

**DOI:** 10.3390/ijms25010051

**Published:** 2023-12-19

**Authors:** Cristina Velázquez-Ruiz, Luis Blanco, María Isabel Martínez-Jiménez

**Affiliations:** Centro de Biología Molecular Severo Ochoa, (CSIC-UAM), c/Nicolás Cabrera 1, Cantoblanco, 28049 Madrid, Spain; cvelazquez@cbm.csic.es

**Keywords:** AEP, human PrimPol, AEP motif B, 3′site dNTP ligands, TLS

## Abstract

PrimPol is a DNA primase/polymerase from the Archaeo-Eukaryotic Primase (AEP) superfamily that enables the progression of stalled replication forks by synthesizing DNA primers ahead of blocking lesions or abnormal structures in the ssDNA template. PrimPol’s active site is formed by three AEP-conserved motifs: A, B and C. Motifs A and C of human PrimPol (*Hs*PrimPol) harbor the catalytic residues (Asp^114^, Glu^116^, Asp^280^) acting as metal ligands, whereas motif B includes highly conserved residues (Lys^165^, Ser^167^ and His^169^), which are postulated to stabilize 3′ incoming deoxynucleotides (dNTPs). Additionally, other putative nucleotide ligands are situated close to motif C: Lys^297^, almost invariant in the whole AEP superfamily, and Lys^300^, specifically conserved in eukaryotic PrimPols. Here, we demonstrate that His^169^ is absolutely essential for 3′dNTP binding and, hence, for both primase and polymerase activities of *Hs*PrimPol, whereas Ser^167^ and Lys^297^ are crucial for the dimer synthesis initiation step during priming, but dispensable for subsequent dNTP incorporation on growing primers. Conversely, the elimination of Lys^165^ does not affect the overall primase function; however, it is required for damage avoidance via primer–template realignments. Finally, Lys^300^ is identified as an extra anchor residue to stabilize the 3′ incoming dNTP. Collectively, these results demonstrate that individual ligands modulate the stabilization of 3′ incoming dNTPs to optimize DNA primer synthesis efficiency during initiation and primer maturation.

## 1. Introduction

Primases have a very important role in DNA replication since they are the only enzymes that can synthetize de novo short primers on an ssDNA template, serving as an elongation point for replicative DNA polymerases. Primases can be divided in two different superfamilies based on their origin and structure: DnaG primases and Archaeo-Eukaryotic Primases (AEPs) [1,2].

*Hs*PrimPol belongs to the AEP superfamily, and is the second primase described in humans [3,4,5]. PrimPols are non-conventional primases that have in common the use of dNTPs for the elongation site or 3′site. At the initiation site or 5′site, they differ in the preferred nucleotide: an NTP is exclusively required by the PrimPol encoded in pRNI (Orf904) from the archaea *Sulfolobus islandicus* [6,7], and a dNTP is strongly preferred in some viral and bacterial PrimPols [8,9,10]; however, *Hs*PrimPol can use both NTPs and dNTPs at the 5′ initiation site [2,11]. The main role of *Hs*PrimPol is to rescue replication forks that have been stalled due to replicative stress, both in nucleus and mitochondria, by providing DNA primers to the replicative polymerase [3,4,12,13]. As depicted in Figure 1A, *Hs*PrimPol is constituted by an AEP core, a Zinc Finger domain (ZnFD), and an RPA-binding domain (RBD). The RBD, located at the very C-terminal region of *Hs*PrimPol [4,14,15,16], is only required in vivo and regulates the recruitment of *Hs*PrimPol at the stalled replication fork [4]. The ZnFD, also located at the C-terminal region preceding the RBD, is indispensable for *Hs*PrimPol primase activities both in vivo and in vitro [4,11,12,17]. The AEP catalytic core contains three conserved motifs: A (DxD/E), B (sxH/R) and C (D/E), which are spatially close at a hydrogen bond distance, suggesting an interaction among these amino acids [18] to form the catalytic active site. Asp^114^, Glu^116^ and Asp^280^ are the invariant carboxylate residues found at motifs A and C of *Hs*PrimPol that interact with the two metal ions required for catalysis [3,4,19] (Figure 1A). Unlike other primases from the AEP superfamily, eukaryotic PrimPols have a Glu as the second carboxylate in motif A (Figure 1C); in *Hs*PrimPol, Glu^116^ seems to be the selector of manganese as the preferred metal ion [19] that allows a faster reaction rate [20]. Mn^2+^ favors the error-prone incorporation of dATP opposite an 8-oxoG lesion, and enables template dislocation reactions that drive *Hs*PrimPol Trans-Lesion Synthesis (TLS) activities [19,21]. Motif B configures the elongation site (3′site) to bind incoming dNTPs.

The crystal structure of *Hs*PrimPol suggests that some residues located within motif B (Ser^160^, Lys^165^, Ser^167^, His^169^) are needed to stabilize the incoming 3′ dNTP [22] (Figure 1B); moreover, other amino acids found within the AEP core were also proposed to contact the 3′ incoming dNTP [22], such as Tyr^100^, Asp^114^, Glu^116^ or Arg^288^, Asn^289^, Arg^291^ and even Lys^297^ and Lys^300^, located beyond motif C (Figure 1B). Among them, Tyr^100^ was described as the 3′ site nucleotide sugar selector that allows *Hs*PrimPol to prefer dNTPs over *NTPs* [23]; Wan and co-workers observed that the H169N mutant had neither primase nor polymerase activities, and this mutant could not rescue cells in vivo that had been exposed to replication stress agents [4]. Arg^291^ was shown to be necessary for PrimPol activities as it stabilizes the 3′ incoming dNTP by contacting the β- and γ-phosphates [24], and it also seems to contribute to the preferential binding of dNTPs over dideoxynucleoside triphosphates (ddNTPs) [24,25]. In addition to these direct ligands, there are other residues that indirectly contribute to the stabilization of the 3′ incoming dNTP by *Hs*PrimPol. For example, the interaction network maintained by both Trp^87^ and Tyr^90^ induces a torsion of β-strand 11, so that Arg^291^ and other amino acids such as Lys^297^ and Lys^300^ are able to contact the 3′ incoming dNTP [26].

In this work, we explored why *Hs*PrimPol has a large number of potential 3′ incoming dNTP-interacting residues, and whether some of them have specific functions during DNA primer synthesis. For that, we studied the role of each of the conserved residues found within motif B of *Hs*PrimPol, and that of residues Lys^297^ and Lys^300^ located close to motif C. Our results demonstrated that His^169^ is essential for dNTP binding, explaining its crucial role for both primase and polymerase activities, whereas Ser^167^ and Lys^297^ are specifically required to synthesize the initial dimer, but not for further primer extension. Interestingly, Lys^165^ was necessary when priming requires the TLS activity of *Hs*PrimPol. Finally, Lys^300^ appears to be an extra anchor residue to stabilize the 3′ incoming dNTP during challenging situations.

## 2. Results and Discussion

### 2.1. Specific Residues at Motif B and Close to Motif C Define a Highly Conserved 3′dNTP Binding Site in Eukaryotic PrimPols

To study the conservation of the different 3′ incoming dNTP ligands of *Hs*PrimPol, a multiple sequence alignment of motifs A, B and C of five groups of the AEP superfamily, including members from animals, plants, archaea, bacteria and phages, was performed (Figure 1C). A preserved c+xsxH consensus within the described motif B could be inferred, where c+ is a positively charged amino acid (Arg, His, or Lys) and **s** is a small amino acid (Ser or Gly) [18]. The aforementioned H (histidine) is invariant within the alignment, suggesting its relevance also for PrimPols. In *Hs*PrimPol, the residues forming motif B are Lys^165^, Ser^167^ and His^169^ [3]. According to the multiple alignment, Lys^297^ and Lys^300^ (close to conserved motif C) were also conserved, specially Lys^297^, which is almost invariant (Figure 1C). The crystal structure of *Hs*PrimPol [22] confirmed that Lys^165^, Ser^167^, His^169^, Lys^297^ and Lys^300^ are spatially close, and that they could work as 3′ incoming dNTP ligands (Figure 1B). To study their individual role in 3′ incoming dNTP stabilization, the alanine substitution of four of these residues (K165A, S167A, K297A and K300A) was selected to preclude the multiple contacts that these amino acids can establish with the 3′ incoming dNTP. His^169^ has been previously investigated, as Wan and co-workers observed that the H169N mutant had neither primase nor polymerase activities [3], but its precise role in dNTP binding was not determined. Therefore, two different His^169^ mutants were included in our study: a change to an alanine (H169A) to abolish any interaction of the side chain, or to a tyrosine (H169Y) to maintain the aromatic ring and its potential function. The different mutant PrimPols were expressed and purified as described in the Materials and Methods section.

### 2.2. Lys^165^, Ser^167^, His^169^ and Lys^297^ Are Essential to Stabilize the 3′ Incoming dNTP in a Preternary Complex with ssDNA

The mechanism of DNA primer synthesis by *Hs*PrimPol involves a number of consecutive steps, including the binding of ssDNA, metals and nucleotides, and the conformational changes required for the initiation of primer synthesis and further maturation [11,27,28]. Thus, *Hs*PrimPol (a scheme of the AEP core and the ZnFD is shown in Figure 2A(a)) firstly forms a binary complex (enzyme:ssDNA) with ssDNA, without metal requirements or the involvement of the ZnFD [11] (Figure 2A(b)). Firstly, the capacity to form an enzyme:ssDNA binary complex was measured in all *Hs*PrimPol mutants using an EMSA assay, as described [11]. As shown in Figure 2B, this binary complex migrates more slowly compared to the free ssDNA in a native polyacrylamide gel, producing a retarded band. It is worth noting that any change in the electrophoretic mobility of the binary complex formed with the mutant PrimPols could reflect an alteration in their tridimensional structure, or a significant change in the interaction with ssDNA. Figure 2B,C shows that all mutants formed an enzyme:ssDNA complex similar to the WT in both mobility and in the enzyme concentration required, meaning that the ssDNA-binding affinity of these mutants was not altered. This also demonstrates that none of the mutants (K165A, S167A, H169A, H169Y, K297A and K300A) have a remarkably altered structural conformation. Moreover, these results underline that motif B and motif C are probably not required for ssDNA binding, in agreement with Rechkoblit and coworkers [22], who proposed that the Module N-terminal of the catalytic core (ModN) is responsible for the main template interactions, a proposal supported in further work [29].

One of the main characteristics of *Hs*PrimPol is that it makes DNA primers instead of conventional RNA primers [3]. To do so, this enzyme must have a particular configuration of the catalytic center that favors the stabilization of a 3′ dNTP, probably due to specific ligands for the 3′ incoming dNTP. Thus, the next step in the primase reaction is the metal-dependent binding of a 3′ incoming dNTP (Figure 2A(c)), establishing the preternary complex (enzyme:ssDNA:dNTP) [1,11]. This step is also independent of the presence of the ZnFD [11]. To evaluate the formation of the preternary complex, the same 60-mer oligonucleotide containing the preferred priming sequence “GTC”, described previously in the analysis of the binary complex, was used in combination with labeled [α-^32^P]dGTP as the 3′ incoming dNTP (complementary to the C of the “GTC” sequence). Figure 2D shows that the K300A mutant was able to form a preternary complex, but less efficiently compared to WT *Hs*PrimPol. Conversely, K165A, S167A, K297A and H169A/Y mutants were unable to bind [α-^32^P]dGTP at the 3′ site in the presence of ssDNA, demonstrating their direct role in the template-directed stabilization of the 3′ dNTP. Interestingly, a different binary complex (enzyme:dNTP) can be also observed with WT *Hs*PrimPol when using labeled [α-^32^P]dGTP, in agreement with previous reports describing that some primases can also bind a 3′ dNTP prior to ssDNA binding [1,30,31,32]. Similarly, as described above for the preternary complex, the K300A mutant could form the enzyme:dNTP binary complex, but less efficiently compared to WT (Figure 2D). This result further emphasizes the direct role of Lys^165^, Ser^167^, His^169^ and Lys^297^ in 3′ dNTP binding at the elongation site of *Hs*PrimPol.

### 2.3. His^169^ Is Crucial, Ser^167^ and Lys^297^ Are Relevant, While K^165^ Is Dispensable and K^300^ Is Irrelevant for De Novo Dimer Synthesis

Once the pre-ternary complex (enzyme:ssDNA:dNTP) has been formed, a ribonucleoside triphosphate can occupy the neighbor 5′ site of *Hs*PrimPol, via base pairing with the template, prior to catalysis of the initial dimer coupled to PPi release (Figure 2A(d); scheme at Figure 3A). Thus, a labeled band corresponding to the *_3p_A*dG dimer can be formed by WT *Hs*PrimPol when providing the preferred template sequence “GTC” and a combination of increasing concentrations of *ATP* as the 5′ nucleotide and [α-^32^P]dGTP as the limiting 3′ incoming dNTP (Figure 3A). As expected from its null capacity to form a pre-ternary complex, both His^169^ mutants were completely inactive to form a *_3p_A*dG dimer even at the highest *ATP* concentration used (Figure 3A). S167A and K297A mutants, strongly affected in the formation of the pre-ternary complex, were also very inefficient in dimer synthesis, and only at the highest concentration of *ATP* a weak band of *_3p_A*dG was observed (Figure 3A). K165A mutant, also affected in the formation of the pre-ternary complex, had a lower production of *_3p_A*dG dimers (72, 60 and 32% at each *ATP* concentration tested; quantification of three independent experiments were represented in Figure 3B). In good agreement with its capacity to form a pre-ternary complex, mutant K300A was competent to produce *_3p_A*dG dimers, even slightly better than the WT *Hs*PrimPol (around 110% as average of the three *ATP* concentrations; Figure 3A,B).

### 2.4. Lys^165^, Ser^167^ and Lys^297^ Are Dispensable for Dimer Extension, but Facilitate Formation of Non-Canonical Extension Products, and His^169^ Is Essential also for Primer Extension

As shown before in this study, all *Hs*PrimPol mutants tested can bind ssDNA, but they are severely affected in stabilizing the 3′ incoming dNTP (except for K300A mutant). Accordingly, these defective mutants showed a total or partial loss of the ability to form nucleotide dimers. To further evaluate the residual capacity of the putative mutants in the 3′dNTP ligands to make DNA primers, a more physiological concentration (100 µM) of 3′ incoming dNTPs was used. Primers were labeled using *[γ-^32^P]ATP* (16 nM) as 5′ site labeled nucleotide, and a 60-mer ssDNA oligonucleotide containing the preferred priming sequence “GTC” in a 10-mer heteropolymeric sequence flanked by a poly T, was used as template. Addition of one or more dNTPs (dGTP, dTTP and dCTP) allows the progressive formation of the primer, step by step (*_3p_A*dG dimer, *_3p_A*dGdT trimer…) until its complete extension (see scheme at Figure 4A). As shown in the autoradiogram, WT *Hs*PrimPol could efficiently synthetize a dimer and extend it to the final product of 10-mer but also yielded additional bands (indicated by * in Figure 4A), that were previously described as non-canonical products [33]. These non-canonical products are the result of primer realignments and template distortions that enable *Hs*PrimPol to continue primer elongation when the next nucleotide to be incorporated is absent.

As shown in Figure 4A, K300A mutant produced a primer extension pattern very similar to the WT, including the non-canonical products, indicating again its WT-like phenotype. In this situation where the concentration of the 3′ incoming dNTP is not restrictive, K165A mutant synthetized a similar amount of dimers compared to the WT *Hs*PrimPol. Nevertheless, K165A mutant dimers were extended more faithfully producing only canonical products based on the nucleotide availability and up to 10-mer products; as mentioned, WT *Hs*PrimPol could yield non-canonical products, even longer than the expected 10-mer which were generated by slippage. These results indicate that Lys^165^ is necessary to yield non-canonical products.

As also shown in Figure 4A, H189A and H169Y were completely inactive in forming dimers, even at the higher concentration (100 µM) of the 3′ incoming dGTP. Moreover, S167A and K297A mutants synthetized dimers very poorly, as observed in the previous experiment (Figure 3), even under these favorable conditions, but, interestingly, this low number of dimers appeared to be fully extended (some 10-mer products are observed). These results suggest that both Ser^167^ and Lys^297^ are only specifically involved in the stabilization of the 3′ incoming dNTP during dimer synthesis.

Next, we further explored whether these 3′ incoming dNTP ligands are important during all nucleotide incorporation cycles, or if their importance is restricted to the initial steps of priming (preternary complex leading to dimer formation), described as the bottleneck of the priming reaction [1,11]. To avoid the need for de novo dimer synthesis, a synthetic mini-primer with three phosphates in the initial ribonucleotide (*_3p_A*dGdT) was used as an ab initio substrate [11], thus mimicking a natural *Hs*PrimPol nascent primer. This pre-formed trimer was used in combination with the 60-mer heteropolymeric template previously described, and elongated with dTTP, dCTP and [α-^32^P]-labeled dGTP (see the scheme shown in Figure 4B). As expected, WT *Hs*PrimPol elongated *_3p_A*GT efficiently up to the end of the heterologous sequence template (10-mer) preceding the polyT tail (dA is not provided), but also producing longer non-canonical products likely due to backwards primer realignments (Figure 4B). When K165A and K300A mutants were analyzed, they also yielded *_3p_A*GT-elongated products, in amounts similar to those produced by WT *Hs*PrimPol. It is worth mentioning that the K165A mutant again produced a reduced number of non-canonical products. As predicted, the S167A and K297A mutants were able to extend a significant amount of the pre-formed *_3p_A*GT trimers, but the non-canonical products were also decreased in these two mutants. Therefore, the initial synthesis of dimers appears to be the step most affected in both S167A and K297A mutants, implying that Ser^167^ and Lys^297^ are especially relevant in stabilizing the preternary complex (enzyme:ssDNA:dNTP), which precedes the binding of the 5′ nucleotide and subsequent dimer synthesis. The altered balance of canonical vs. non-canonical products synthesized by K165A, S167A and K297A mutants compared to the WT PrimPol suggests that Lys^165^, Ser^167^ and Lys^297^ are required for primer realignment-mediated dNTP incorporation [12,34], enabling PrimPol to carry out TLS primase activities [33]. Interestingly, neither H169A nor H169Y showed any extension of the trimer (*_3p_A*GT), leading to the conclusion that His^169^ is also crucial for 3′ dNTP binding during primer elongation.

We propose that the binding of the 3′ incoming dNTP needs residues of Ser^167^, His^169^ and Lys^297^ to provide the maximal stabilization at the active center to form the preternary complex, and to be maintained until the initiating dimer is formed. Further primer extension facilitates the stability of the primer, establishing potential new interactions with PrimPol and, consequently, alleviating the need for a strong interaction with incoming dNTPs. This could explain why Ser^167^ and Lys^297^ are needed for the binding of the first dNTP (dimer formation), but not for binding the next incoming dNTPs. Conversely, His^169^ is indispensable for elongation, probably because it is a main anchor of the incoming dNTP.

### 2.5. Lys^165^, Ser^167^ and Lys^297^ Are Necessary to Skip Unreadable Lesions

*Hs*PrimPol has the capacity to tolerate lesions during DNA priming, either by the direct reading of non-bulky lesions such as 8-oxoG, or by inducing primer realignments to skip unreadable lesions such as abasic sites [34]. Therefore, assuming that the reinforcement/modulation of 3′dNTP binding could be beneficial for the TLS abilities of *Hs*PrimPol, we tested whether the K165A, S167A, K297A and K300A mutants were affected during the TLS events occurring during DNA priming. For that, we used a variation of the 60-mer ssDNA template (3′-T_29_-G**TC**AXACAGCA-T_20_-5′) in which the X base (located at the fourth position of the template relative to the dimer formation site TC, shown in bold) can be either a normal dG, an 8-oxoG lesion, or an abasic site (Ab). As shown in Appendix A, WT *Hs*PrimPol and mutants K165A, S167A, K297A and K300A used both dG and 8-oxoG templates equally well, showing an identical pattern of priming as that shown in Figure 4A, thus indicating that none of these residues is specifically required to tolerate an 8oxoG lesion.

To evaluate the priming competence of mutants K165A, S167A, K297A and K300A when confronting unreadable template lesions, we used the preferred 3′GTC template, but with either a normal dG (Figure 5A) or an abasic (Ab) site (Figure 5B) in the fourth nucleotide to be copied. Again, we took advantage of providing the synthetic 3-mer primer (*_3p_A*GT) to facilitate the TLS analysis of mutants with a severe defect in the initial step of priming. As shown in the upper schemes (Figure 5A,B), PrimPol is able to locate this *_3p_A*GT primer opposite a 3′TCA complementary sequence in the template, next to the cryptic G of the 3′GTC preferred priming site [11], thus ready for a next dNTP insertion. To dissect the different options to read or skip the Ab site, limited combinations of dNTPs were used, as indicated in Figure 5, but providing [α-^32^P]dGTP to label the extension products in all cases.

In the non-damaged template (Figure 5A), and using only [α-^32^P]dGTP, the WT *Hs*PrimPol generated a _3p_AGT*G product that likely corresponds to the insertion of [α-^32^P]dGTP opposite the next dC in the template; this event requires the extrusion of two nucleotides (AG) in the template and the realignment of the 3′T of the *3pA*GT primer opposite the next available A in the template (lane 2, red asterisk; red lettering in the scheme below the autoradiogram). The same event, leading to the formation of *_3p_A*GT*G product, was also observed in the template with the Ab site, where WT *Hs*PrimPol dislocated AAb in the template, thus skipping the lesion (Figure 5B, lane 22). The K300A mutant was able to carry out similar WT-like reactions in the non-damaged template and in the one containing the Ab site (Figure 5A,B lanes 18 and 38, respectively). However, the K165A, S167A and K297A mutants were unable to produce any product in these conditions, therefore confirming previous results indicating that they cannot promote primer realignment-dependent insertions of dNTPs in either of the two templates (Figure 5A, lanes 6, 10 and 14; Figure 5B, lanes 26, 30, 34).

When [α-^32^P]dGTP was supplemented with dATP, the WT *Hs*PrimPol, K165A and K300A mutants efficiently formed a de novo _3p_A*G dimer in both non-damaged and damaged templates (Figure 5A, lanes 3, 7 and 19; Figure 5B, lanes 23, 27 and 39; cyan asterisk and lettering in the schemes). As expected, the S167A and R297A mutants produced limited _3p_A*G products in both damaged and undamaged templates (Figure 5A, lanes 11 and 15; Figure 5B, lanes 31 and 35) due to their low capacity for dimer synthesis.

When [α-^32^P]dGTP was supplemented with dCTP, the WT *Hs*PrimPol, K165A and K300A mutants gave the *_3p_A*GTC*G product, but only in the non-damaged template (Figure 5A, lanes 4, 8 and 20, respectively; pink asterisk and lettering in the scheme), explaining that direct dC extension cannot occur opposite the Ab site. The insertion of the labeled dG can occur in two ways: (1) after TC realignment coupled to AGAC extrusion; (2) after the extrusion of the next template A (see schemes shown in Figure 5A). Interestingly, whereas the first means of insertion (involving primer realignment) is predominant for WT *Hs*PrimPol, the second means of insertion is predominant with the K165, S167 and K297 mutants, as evidenced by the appearance of longer extension products (indicated by purple asterisks in Figure 5A, corresponding to the purple lettering products in the schemes). In the damaged (Ab) template, the WT and K300A mutant produced *_3p_A*GT*G as a consequence of primer realignment and the extrusion of AbA, and _3p_AGT*GC by the further extrusion of A and dC insertion (Figure 5B, lanes 24 and 40, red and dark blue asterisks; see also the schemes). In agreement with their primer realignment disability, K165A, S167A and K297A mutants lack this solution to skip the Ab site (Figure 5B, lanes 28, 32 and 36).

Finally, when [α-^32^P]dGTP was supplemented with dTTP, newly synthesized _3p_GT dimers could only be observed with WT *Hs*PrimPol, K165A and K300A mutants in both templates (Figure 5A, lanes 5, 9 and 21; Figure 5B, lanes 25, 29 and 41), which are the ones competent in dimer synthesis, as shown previously. In addition, both WT *Hs*PrimPol and the K300A mutant gave rise to *_3p_A*GTT*GT as a consequence of primer realignment and AG extrusion before nucleotide insertion, and further elongated products (see green lettering and asterisks in Figure 5A, lanes 5 and 21, respectively). These products also appeared when the template, instead of the G, harbored an easier-to-extrude Ab site (Figure 5B, lanes 25 and 41; see also the schemes). Once more, the K165A, S167A and K297A mutants gave no elongation products in the non-damaged template (Figure 5A, lanes 9, 13 and 17, respectively) or in the damaged template (Figure 5B, right panel, lanes 29, 33 and 37, respectively).

From all of these results, it can be concluded that Lys^165^, not essential to the formation of dimers, is specifically required to skip unreadable lesions during priming by allowing primer realignment-driven dNTP insertions via *Hs*PrimPol. It is likely that Lys^165^ reinforces dNTP binding (perhaps strengthening the interaction with the phosphates) in those situations when the templating base does not provide the optimal base-pairing stability to the complementary dNTP, as it luckily occurs during damage avoidance. Ser^167^ and Lys^297^ are more critical in stabilizing the 3′ incoming dNTP during dimer synthesis, which is the bottleneck of the priming reaction [1,2,3,4,5,6,7,8,9,10,11], but they seem to play a role in posterior steps, i.e., dNTP insertions that facilitate lesion skipping. Therefore, these specific residues of *Hs*PrimPol can serve to modulate the binding of the 3′ incoming dNTP, reinforcing it during reactions involving primer realignments or template distortions, somehow compensating the limited interactions of *Hs*PrimPol with the DNA template [22].

### 2.6. His^169^ Is Essential for DNA Polymerase Activity. Lys^165^, Ser^167^ and Lys^297^ Are Also Important for Stabilizing the 3′ Incoming dNTP in a Conventional DNA Polymerase Assay

The defects in primer synthesis observed with these mutants of putative dNTP ligands could be due to two main reasons that are not mutually exclusive: a) the resulting dNTP binding affinity is reduced and/or b) there is a defective catalysis due to a misorientation of the 3′ incoming dNTP. To clarify this, the kinetic parameters (Km, kcat) of nucleotide insertion by these mutants were calculated in a reaction that measures a +1 nt insertion, using a standard DNA polymerase assay where the substrate is a 28-mer DNA template annealed to a labeled 15-mer primer. But first, it was necessary to check that these *Hs*PrimPol mutants conserved some DNA polymerase capacity. To do so, a conventional DNA polymerase assay was carried out where the primer is extended up to the end of the template DNA when increasing concentrations of dNTPs are provided. As observed in Figure 6A, PrimPol WT could extend the labeled primer up to the end of the template with 1 µM dNTP concentration. As expected, both H169A and H169Y mutants had no detectable DNA polymerase activity as no primer elongation was observed, even at 100 µM dNTPs, showing again that His^169^ is indispensable at each cycle of dNTP binding at the 3′ site. In congruence with our primase assays, S167A and K297A mutants could elongate the pre-existing DNA primer to some extent when using high dNTP concentrations (10–100 μM), showing inefficient polymerase activity compared to the WT *Hs*PrimPol at 1 µM dNTP (Figure 6B). Similarly, mutant K165A could extend the DNA primer completely, but using dNTP concentrations 10–100-fold higher than the WT to achieve a similar level of primer extension (Figure 6A,B). Finally, mutant K300A was able to elongate the primer at 1 µM dNTP (Figure 6B), although it was less competent than WT in completing the extension up to the end of the template (fewer primers could be extended more than 7 nt compared to the WT; Figure 6A).

Once it was confirmed that these mutants conserved the DNA polymerase activity (except for His^169^ mutants), the kinetic parameters (Km, kcat) in steady-state conditions were calculated for each individual dNTP +1 addition opposite each complementary templating base (Table 1). Compared to WT *Hs*PrimPol, the K165A mutant showed an increased Km corresponding to a 5- to 20-fold reduction in dNTP binding affinity (considering the four different dNTPs), whereas its catalytic rate (kcat) remained almost similar or decreased around twofold. Consequently, the relative activity of K165A compared to the WT *Hs*PrimPol, estimated from the ratio of catalytic efficiencies (kcat/Km(mutant)*100/kcat/Km(WT), gave an averaged value of 7.2%. The dNTP affinity of the S167A mutant largely decreased (70–500-fold), and its catalytic rate also reduced about threefold; in the case of the K297A mutant, its dNTP affinity was also strongly reduced (40–330-fold), but less drastically than for the S167A mutant, whereas its catalytic rate was more significantly reduced (five- to ninefold) compared to WT *Hs*PrimPol. These results imply a residual relative activity of 0.26% for S167A, and 0.18% for K297A. The dNTP binding affinity of the K300A mutant remained similar for dATP and dTTP and slightly higher for dGTP, but it was reduced sevenfold for dCTP, suggesting it to be a specific ligand of this small dNTP. The K300A catalytic rate was reduced about twofold for dGTP, but was similar for the other dNTPs. As a result, the relative activity for K300A can be calculated as 44% on average, and the individual effects observed for specific dNTPs could partially explain the limited elongation pattern shown in Figure 6A. These results demonstrate that these residues are not only important for the template-dependent binding of the 3′ incoming dNTP, but also for its orientation in a productive catalytic configuration.

### 2.7. Structural Modeling of Mutations and Inferred Function for the dNTP Ligands of HsPrimPol

In order to understand the structural bases of each studied amino acid in detail, we took advantage of the *Hs*PrimPol core crystal structure and modeled the interaction rotamers of the WT and the mutated amino acid of interest.

**Lys^165^**: *Hs*PrimPol’s crystal structure [22] shows that Lys^165^ can make (1) hydrogen bonds with the oxygens of the γ-phosphate of the 3′ incoming dNTP, and (2) steric clashes with the metal ligand Glu^116^ (Figure 7A(a–c)). Likewise, some rotamers of Glu^116^ establish this type of negative contact with Lys^165^. The substitution of the lysine by an alanine (K165A) as K297A cannot maintain these interactions (Figure 7A(d)), which causes reduced binding and the altered positioning of the 3′ incoming dNTP in the catalytic center, explaining the reduction in both the dNTP affinity and the catalytic rate of this mutant. *Hs*PrimPol, which maintains few contacts with the DNA template [22], probably requires a tight binding of the 3′ incoming dNTP to carry out any insertion associated with primer–template realignments, again explaining the importance of Lys^165^ in this scenario. Moreover, the catalytic residue Glu^116^, with which Lys^165^ maintains steric clashes, must be correctly positioned to bind Mn^2+^. This metal ion enables PrimPol to adopt an optimal catalytic core conformation to stabilize the 3′ incoming dNTP during TLS primase activities [19]. Then, any alteration of this metal adjustment would also hinder PrimPol TLS primase activities, as it occurs when Lys^165^ is not present. In conclusion, *Hs*PrimPol residue Lys^165^, conserved in the AEP superfamily and located within motif B, is needed to perform TLS primase activities, likely contributing to the tighter binding of the 3′ incoming dNTP when primer–template realignments are required to bypass lesions. Lys^165^ also positions the catalytic residue Glu^116^ to maintain interactions with Mn^2+^ as metal cofactor, which enables the acquisition of an optimal catalytic conformation.

**Ser^167^**: The crystal structure of *Hs*PrimPol [22] shows that Ser^167^ can form hydrogen bonds with (1) the oxygen of the γ-phosphate of the incoming dNTP; (2) with Ser^160^, which, in turn, also establishes hydrogen bonds with one oxygen atom of the γ-phosphate; and (3) with the metal ligand Glu^116^ (Figure 7B(a–c)). As previously mentioned, Glu^116^ coordinates the metal ion B involved in the two-metal-ion mechanism, allowing catalysis [19,35,36]. In fact, the metals, the catalytic residues and the nucleotidic substrate need to be perfectly aligned so that catalysis can take place [37]. The substitution of Ser^167^ with an alanine (S167A) causes the loss of the interactions with the γ-phosphate of the 3′ incoming dNTP, the side chain of Ser^160^ and the Glu^116^ (Figure 7B(d)). It has been shown that the S167A mutant is especially defective in dimer formation, which is the bottleneck of the initiation process in primases [1]. The kinetic parameters of the S167A mutant during primer elongation revealed that its dNTP affinity was severely reduced compared to the WT (70–500-fold) and the catalytic rate dropped by around 30% (Table 1). In conclusion, *Hs*PrimPol residue Ser^167^, invariant within motif B in the Eukaryotic PrimPol group, could have two different roles that are not mutually exclusive: (1) stabilizing and orientating the 3′ incoming dNTP, either directly or indirectly through the proper positioning of Ser^160^; and (2) locating the catalytic Glu^116^ in an optimal configuration for catalysis. Both functions would be especially required during two primer synthesis scenarios: dimer synthesis and dNTP insertions associated with TLS.

**His^169^**: The crystal structure of *Hs*PrimPol [22] shows that His^169^ can form (1) hydrogen bonds with the catalytic Asp^114^ within motif A, and (2) direct interactions with the oxygen of the β-phosphate of the 3′ incoming dNTP (Figure 7C(a–b)). When the His^169^ of *Hs*PrimPol is substituted with an alanine (H169A), all of these interactions are lost (Figure 7C(c)). If the histidine is changed to a tyrosine (H169Y), steric clashes appear with the oxygens of the β-phosphate and the proper β-phosphate (Figure 7C(d)); moreover, in the H169Y mutant, a hydrogen bond is formed with the oxygen that lies between the γ- and β-phosphates and a rotamer of the tyrosine can then form steric clashes with Asp^114^ instead of hydrogen bonds (Figure 7C(d)). As shown here and in recent work [38], the lack of His^169^ severely affected 3′ dNTP binding; however, further reasons could explain the complete inactivity of H169A and H169Y mutants: a) Asp^114^ could adopt an unproductive configuration due to the loss of the hydrogen bond with His^169^, and/or b) the disappearance of the His^169^ interactions could result in an improper orientation of the 3′ incoming dNTP. A different mutation H169N, studied by Wan and coworkers, was unable to synthetize primers and, hence, to complement PrimPol-deficient cells [4]. When this mutation was modeled, the asparagine could still form hydrogen bonds with the Asp^114^, but no interactions with the dNTP could be inferred. Altogether, this indicates that the primary role of His^169^ is barely the stabilization of the 3′ incoming dNTP, binding and likely orientating its phosphates at the proper distance to receive the nucleophilic attack, thus being absolutely necessary in every single step of primer synthesis. Nevertheless, a contribution of His^169^ in the orientation of the Asp^114^ cannot be completely ruled out. The conservation of this His within motif B of the whole AEP superfamily agrees with its critical role in conventional p48 primase and in other PrimPols such as Orf904 from *Sulfolobus islandicus*, deep sea phages and Mimivirus [6,10,39,40].

**Lys^297^**: The crystal structure of *Hs*PrimPol [22] shows that Lys^297^ can form (1) hydrogen bonds with the α-, β- and γ-phosphates of the 3′ incoming dNTP and (2) direct interactions with Lys^300^ (Figure 7D(a–c)). Interestingly, some rotamers of Lys^297^ can establish steric clashes with the oxygen between the β- and γ-phosphates or maintain negative interactions with Lys^300^. The substitution of Lys^297^ to alanine (K297A) only allows positive interactions with Lys^300^, but abolishes any interaction with the incoming dNTP (Figure 7D(d)), explaining its reduced dNTP affinity and decreased ability to synthesize dimers. Strikingly, mutant K297A was still able to elongate synthetic mini-primers, suggesting that Lys^297^ is dispensable during further steps of dNTP insertion for primer maturation. Moreover, as it can be observed in the crystal structure of *H*sPrimPol [22], Lys^297^ is one of the residues that configure the catalytic pocket. The lack of Lys^297^ likely destabilizes this configuration, also explaining the reduction in the catalytic rate of the K297A mutant. In conclusion, the *Hs*PrimPol residue Lys^297^, invariant in Eukaryotic PrimPols and situated just beyond motif C, is essential for dimer formation, but partially dispensable during subsequent elongation steps. Lys^297^ interacts with multiple elements of the 3′ incoming dNTP, therefore increasing its binding affinity and serving to orientate the dNTP in the catalytic center.

**Lys^300^**: The crystal structure of *Hs*PrimPol [22] shows that Lys^300^ can form (1) hydrogen bonds with Lys^297^ that are dependent on the backbone (Figure 7E(a–c)) and, therefore, are preserved in the K300A mutant (Figure 7E(d)), and (2) either a hydrogen bond with the oxygen of the γ-phosphate of the dNTP (panel b) or an interaction with Arg^291^ (panel c), but these alternative contacts are lost in the K300A mutant (panel d). The K300A mutant exhibited a WT-like phenotype, although its DNA primase and polymerase activities were slightly decreased compared to the WT *Hs*PrimPol. The specific measurement of the kinetic parameters during polymerization activity of K300A mutant showed a slightly reduced dNTP affinity compared to the WT, as well as a small decreased catalytic rate, resulting in a dNTP incorporation efficiency of 50% compared to the WT. Lys^300^ could have a structural role as a foundation of the bridge formed by K297A with K165A that clinches the 3′dNTP in the catalytic pocket (see Figure 1B). In conclusion, Lys^300^ (moderately conserved in the AEP superfamily) has a minor role in 3′dNTP binding, but can contribute to the formation of the preternary complex.

## 3. Materials and Methods

### 3.1. Primary Sequence Alignments, Structure Visualization and Modeling of HsPrimPol Mutants

Multiple sequence alignments were performed using the COBALT (Constraint-based Multiple Alignment Tool) server (https://www.ncbi.nlm.nih.gov/tools/cobalt/cobalt.cgi; accessed on 7 January 2021) from the National Center of Biotechnology Information (NCBI). The modeling of three-dimensional structures was performed with both the Swiss-PdbViewer (DeepView; Version 4.1.1) program (from the Swiss Institute of Bioinformatics (SIB), Lausanne, Swiss) and PyMOL Molecular Graphics System (Version 2.0, Schrödinger, LLC, USA). The crystal structure of the *Hs*PrimPol ternary complex (PDBid: 5L2X) was used for modeling [22].

### 3.2. Mutagenesis and Purification

*Hs*PrimPol point mutations (K165A, S167A, H169A/Y, K297A, K300A) were introduced into expression plasmid pET16::PrimPol [3], following the protocol of the QuickChange II Site-Directed Mutagenesis Kit (Stratagene, San Diego, CA, USA) and using the following oligonucleotides: K165A sense (5′CTTGGATTCTAGCACTGATGAAGCAT TCAGCCGGCATTTAATATTTC 3′); K165A antisense (5′GAAATATTAAATGCCGG CTGAATGCTTCATCAGTGCTAGAATCCAAG 3′); S167A sense (5′CACTGATGAAAA ATTCGCCCGGCATTTAATATTTCAGCTCC 3′); S167A antisense (5′GGAGCTGAAATATTAAATGCCGGGCGAATTTTTCATCAGTG 3′); H169A sense (5′GAAAAATTCAG CCGGGCTTTAATATTTCAGCTCCATGATGTGG3′); H169A antisense (5′CCACATCA TGGAGCTGAAATATTAAAGCCCGGCTGAATTTTTC 3′); H169Y sense (5′GAAAAA TTCAGCCGGTATTTAATATTTCAGCTCCATGATGTGG 3′); H169Y antisense (5′CCA CATCATGGAGCTGAAATATTAAATACCGGCTGAATTTTTC 3′); K297A sense (5′CG GCTATATAAATCATCAGCAATTGGAAAGCGTGTGG 3′); K297A antisense (5′CCA CACGCTTTCCAATTGCTGATGATTTATATAGCCG 3′); K300A sense (5′GCTATATAAATCATCAAAAATTGGAGCGCGTGTGGCTTTGG 3′); and K300A antisense (5′CCAA AGCCACACGCGCTCCAATTTTTGATGATTTATATAGC 3′). The *Hs*PrimPol encoding gene was sequenced to confirmed that only the desired change was introduced. All *Hs*PrimPol mutants were expressed and purified as previously described [3].

### 3.3. Nucleotides and Oligonucleotides

Ultrapure dNTPs/NTPs were supplied by GE (Fairfield, CT, USA). The radiolabeled nucleotides [γ-^32^P]ATP (3000Ci/mmol) and [α-^32^P]dGTP (3000 Ci/mmol) were obtained from Perkin Elmer (Waltham, MA, USA). DNA oligonucleotides were synthesized by Sigma Aldrich (St. Louis, MO, USA) and IDT (Coralville, IO, USA), whereas the *_3p_A*GT mini-primer was purchased from Tebubio (Le Perray-en-Yvelines, France). T4 polynucleotide kinase (New England Biolabs, Ipswich, MA, USA) was used to label the oligonucleotides at the 5′ end.

### 3.4. EMSA for Binary and Preternary Complex Detection

Electrophoretic mobility shift assays (EMSA) were used to evaluate the capacity of different *Hs*PrimPol mutants to bind ssDNA, thus forming an enzyme:ssDNA binary complex. The reactions were performed in buffer A (50 mM Tris-HCl pH 7.5, 25 mM NaCl, 2.5% (*w*/*v*) glycerol, 5% (*w*/*v*) PEG-4000, 1 mM DTT, 0.1 mg/mL BSA) using T_20_-GTCC-T_36_ (0.6 nM) labeled at the 5′ end with PNK and [γ-^32^P]ATP as the template. Increasing concentrations of the WT *Hs*PrimPol or mutant variants (5, 20 and 80 nM) were added to the reactions, and then incubated at 30 °C for 10 min. Loading buffer (30% glycerol, 1 mM EDTA, 0.1% xylene cyanol and 0.1% bromophenol blue) was then added to samples and the reactions were run in a native 6% polyacrylamide gel at 180 V for 120 min at 4 °C in Tris–glycine pH 8.3 buffer. The mobility shift of the enzyme:ssDNA complex versus free ssDNA was analyzed using autoradiography.

Direct binding to the 3′ incoming dNTP by the different *Hs*PrimPol mutants, thus forming an enzyme:dNTP binary complex, was also evaluated using EMSA. The reactions were performed in buffer B (50 mM Tris pH 7.5, 25 mM NaCl, 2.5% (*w*/*v*) glycerol, 1.25% (*w*/*v*) PEG-4000, 1 mM MnCl_2_, 1 mM DTT, 0.1 μg/μL BSA) by adding WT *Hs*PrimPol or mutant variants (400 nM) and [α-^32^P]dGTP (16 nM). The reactions were incubated at 30 °C for 10 min. Then, loading buffer was added and the complexes were analyzed as described before.

The ability to form a preternary complex between different *Hs*PrimPol mutants, ssDNA and 3′ incoming dNTP (enzyme:ssDNA:dNTP), was evaluated using EMSA. The reactions were performed in buffer B (50 mM Tris pH 7.5, 25 mM NaCl, 2.5% (*w*/*v*) glycerol, 1.25% (*w*/*v*) PEG-4000, 1 mM MnCl_2_, 1 mM DTT, 0.1 μg/μL BSA) by adding WT *Hs*PrimPol or mutant variants (400 nM), [α-^32^P]dGTP (16 nM) as the 3′ site dNTP, and a template of 60-mer ssDNA 3′-T_20_-GTCC-T_36_-5′ (1 μM). The reactions were incubated at 30 °C for 10 min. Then, loading buffer was added and the reactions were analyzed as previously described.

### 3.5. DNA Primase Assays

The dimer synthesis capacity of either WT or mutant *Hs*PrimPol variants was evaluated by using a 30-mer oligonucleotide 3′-T_10_-GTCC-T_15_-5′ DNA as the DNA template, supplemented with [α-^32^P]dGTP (16 nM) as the 3′ dNTP, and increasing concentrations of *ATP* (1, 10 and 100 μM) as the 5′site nucleotide. To study the dimer formation and elongation across certain DNA lesions, the 60-mer oligonucleotide 3′-T_29_-GTCAXACAGCA-T_20_-5′ was used (where X can be a guanine (G), an 8-oxoG lesion or an abasic site). *[γ-^32^P]ATP* (16 nM) was used as the 5′ nucleotide, supplemented with dGTP, dTTP and dCTP (100 μM) as indicated in each experiment. Alternatively, the processive elongation of a synthetic (pre-formed) primer, *_3p_A*GT (10 μM), by either WT or mutant *Hs*PrimPol on a 60-mer oligonucleotide template 3′-T_29_-GTCAXACAGCA-T_20_-5′ (where X can be a G or an abasic site) was evaluated by providing [α-^32^P]dGTP (16 nM) and dATP, dCTP or dTTP (10 μM) when indicated.

For all assays, reactions (20 μL) in buffer C (50 mM Tris pH 7.5, 25 mM NaCl, 2.5% (*w*/*v*) glycerol, 1 mM MnCl_2_, 1 mM DTT, 0.1 μg/μL BSA), in the presence of either WT or mutant *Hs*PrimPol variants (100 nM), were incubated for 30 min at 30 °C, and stopped by the addition of formamide buffer. Then, the samples were loaded on 8 M urea-containing 20% polyacrylamide sequencing gel and analyzed using autoradiography.

### 3.6. DNA Polymerization Assays

The DNA polymerase activity of WT *Hs*PrimPol or K165A, S167A, H169A/Y, K297A and K300A mutants was assayed in buffer D (0.10 M phosphate buffer pH 6, 40 mM NaCl, 2.5% (*w*/*v*) glycerol, 1 mM MnCl_2_, 1 mM DTT, 0.1 μg/μL BSA) supplemented with the suitable template primer (see below). The WT and mutant variants (50 nM) were incubated with the indicated concentration of dNTPs for 30 min at 30 °C. The reactions were stopped by adding formamide buffer (25 mM EDTA, 95% *v*/*v* formamide and 0.3% *w*/*v* bromophenol blue and xylene cyanol). Then, the samples were loaded on 8 M urea-containing 20% polyacrylamide sequencing gel and analyzed using autoradiography.

The DNA substrate used to assay conventional DNA polymerization was obtained by hybridizing a 28-mer DNA template strand (3′-CTAGTGTCACTCATGGTCTATGTGAAGA-5′) to a 5′ [^32^P]-labeled 15-mer DNA primer strand (5′-GATCACAGTGAGTAC-3′).

### 3.7. Steady-State DNA Polymerization Assay

The kinetic parameters of dNTP incorporation were obtained by measuring the formation of a +1 primer-extended product by either WT or mutant *Hs*PrimPol variants, under steady-state conditions, using 250 nM of a 28-mer ssDNA template (3′-CTAGTGTCACTCATGXTCTATGTGAAGA-5′), where X stands for each of the four (A, C, G or T) bases, hybridized to a 5′ [^32^P]-labeled 15-mer DNA primer (5′-GATCACAGTGAGTAC-3′). Then, +1-extension reactions either by WT, K165A or K300A mutants (50 nM) were carried out with each dNTP, for 30 min at 30 °C, as previously described. In the case of both S167A (50 nM) and K297A (100 nM) variants, the incubation time was increased to 1 h.

## 4. Conclusions

The crystal structure of *Hs*PrimPol shows a variety of putative ligands for 3′ incoming dNTPs located within motif B (Ser^160^, Lys^165^, Ser^167^, His^169^) and within motif C (Arg^288^, Asn^289^, Arg^291^); in addition, Arg^76^, Tyr^100^, Lys^297^ and Lys^300^ also maintain interactions with the 3′ incoming dNTP [22]. In this work, we focused on the role of Lys^165^, Ser^167^, His^169^, Lys^297^ and Lys^300^, showing that the lack of each individual ligand has a different impact on the multiple *Hs*PrimPol capacities. Our structure–function analysis demonstrated that each of these ligands contributes to stabilizing incoming dNTPs, but are differentially important at specific events of the priming process, from forming an initiating dimer to bypassing unreadable lesions during the extension stage of primer synthesis; in addition, all of them participate in the stabilization of the preternary complex, one of the most critical and limiting steps of primer synthesis [1,11]. Collectively, our results suggest that Eukaryotic PrimPols need to have diverse and perhaps alternative ligands to modulate the strength of dNTP binding required to accomplish the de novo initiation and maturation of DNA primers under DNA replication stress conditions.

## Figures and Tables

**Figure 1 ijms-25-00051-f001:**
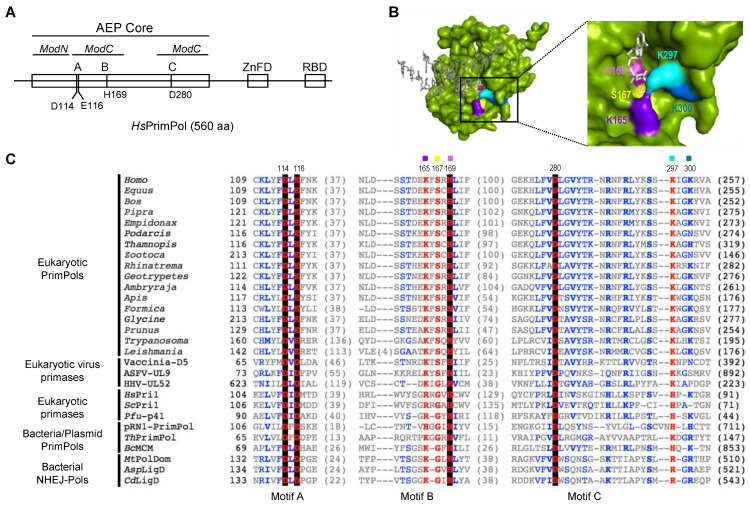
Highly conserved residues at motif B and close to motif C define the 3′ dNTP binding site in the AEP superfamily. (**A**) Scheme of human PrimPol domains. AEP core (in green), subdivided in ModN (aa 35 to 105) and ModC (aa 108–200 and 261–348), as proposed by Rechkoblit et al. (2016) [22]. The AEP core is shared among members of the AEP superfamily, and contains the conserved motifs A (red), B (blue) and C (red). A Zn-finger domain (ZnFD) is shown in purple and the RPA-binding domain (RBD) in orange found at C-terminal region. (**B**) Molecular surface of the AEP core of human PrimPol (green) in preternary complex with the 3′ incoming dNTP (white) and template/primer DNA (gray); as shown in the enlarged area, Lys^165^ (indigo), Ser^167^ (yellow), His^169^ (pink), Lys^297^ (cyan) and Lys^300^ (sky blue) are putative direct ligands of the 3′ incoming dNTP, forming the catalytic pocket together with the three invariant carboxylates (not shown for clarity). (**C**) Amino acid sequence alignment of the three conserved motifs A (shadowed in red), B (shadowed in blue) and C (shadowed in red) of different AEP enzymes, including Eukaryotic PrimPols from *Homo sapiens* (NP_001332824.1), *Equus caballus* (XP_023486679.1), *Bos taurus* (NP_001068956.1), *Pipra filicauda* (XP_027591218.1), *Empidonax traillii* (XP_027735658.1), *Podarcis muralis* (XP_028600037.1), *Thamnophis elegans* (XP_032080602.1), *Zootoca vivipara* (XP_034967959.1), *Rhinatrema bivittatum* (XP_029440727.1), *Geotrypetes seraphini* (XP_033799271.1), *Amblyraja radiata* (XP_032874431.1), *Apis mellifera* (XP_001121815.3), *Formica exsecta* (XP_029670346.1), *Glycine soja* (XP_028181145.1), *Prunus avium* (XP_021827012.1), *Trypanosoma equiperdum* (SCU71797.1), *Leishmania donovani* (XP_003863827.1), D5 primase from *Vaccinia virus* (Vaccinia-D5; P21010.1), UL9 helicase-containing protein from *African Swine Fever Virus* (ASFV-UL9; Q07183), UL52 primase subunit from *Human herpesvirus 1* (HHV-UL52; P10236.1), small subunit of *Homo sapiens* primase (*Hs*Pri1; NP_000937.1), small subunit of *Saccharomyces cerevisae* primase (*Sc*Pri1; P10363.2), catalytic subunit of *Pyrococcus furiosus* primase (*Pfu*-p41; WP_011011222.1), *Sulfolubus islandicus* (pRN1) plasmid pRN1 ORF904 (pRN1-PrimPol; WP_010890202.1), PrimPol from *Thermus thermophilus* (*Th*PrimPol; WP_011173100.1), PrimPol-helicase from *Bacillus cereus* (*Bc*MCM; WP_136983851.1), polymerase domain of *Mycobacterium tuberculosis* ligase D (*Mt*PolDom; WP_003911307.1), ATP-dependent DNA ligase D from *Arthrobacter sp.* (*Asp*LigD; TQK28754.1) and DNA ligase D from *Clostridioides difficile* (*Cd*LigD; VTR08300.1). Non-conserved residues are in grey; conserved amino acids in blue; highly conserved in red and those invariant in black background. Selected *Hs*PrimPol residues are indicated: Asp^114^, Glu^116^ and Asp^280^ (catalytic residues); Lys^165^, Ser^167^ and His^169^ (conserved residues within motif B); Lys^297^ and Lys^300^ (as potential interactors of 3′ incoming dNTPs).

**Figure 2 ijms-25-00051-f002:**
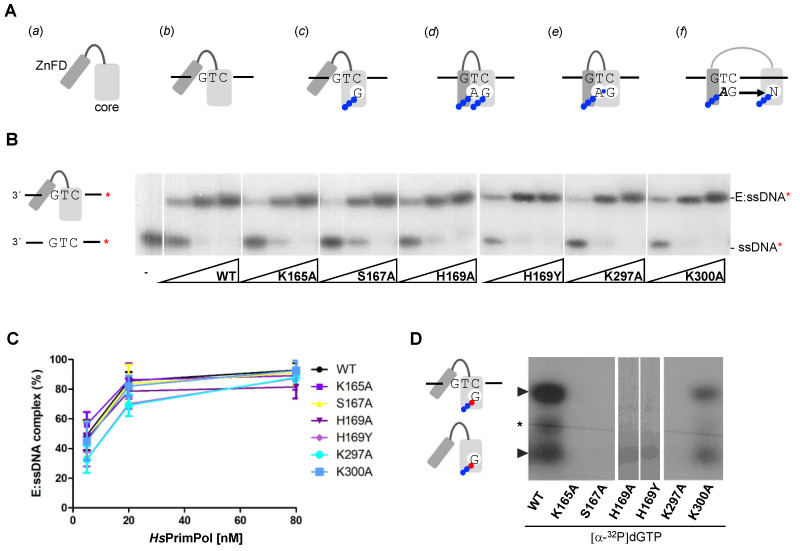
Motif B residues and Lys^297^ are necessary to stabilize the 3′ incoming dNTP before priming. (**A**) Scheme of the mechanism of DNA priming by human PrimPol; (**a**) *Hs*PrimPol modular organization: catalytic core in light gray and Zinc Finger domain in dark gray; (**b**) PrimPol DNA binding, which does not require the ZnFD; (**c**) dNTP (drawn as the base letter and 3 phosphates represented by blue balls) binding at the 3′elongation site that requires manganese ions and pairing with the templating base; (**d**) nucleotide binding at the 5′initiation site, which is configured by both the PrimPol core and the ZnFD; (**e**) dinucleotide formation coupled to PPi release; (**f**) primer elongation, coupled to separation of the PrimPol core and the ZnFD. (**B**) Binary complex (E:ssDNA) detected via EMSA using oligonucleotide 3′ T_20_-GTCC-T_36_ 5′ (0.6 nM; 60-mer; [γ-^32^P]-labeled depicted as a red asterisk) and increasing concentrations of PrimPol WT or mutants (5, 20 and 80 nM). (**C**) Quantification (average of three different experiments) of the E:ssDNA complex using optical densitometry, expressed as percentage of the labeled ssDNA. (**D**) Binary (E:dGTP*) and preternary (E:dGTP*:ssDNA) complexes detected via EMSA using 400 nM of WT or mutants, 16 nM of [α^32^P]dGTP (α^32^P is indicated as a red ball while β and γ -phosphates in blue) and and 1 μM 3′ T_20_-GTCC-T_36_ 5′ (unlabeled) as a template in the presence of Mn^2+^ (1 mM). Non-specific labeling of the oligonucleotide is indicated by a black asterisk.

**Figure 3 ijms-25-00051-f003:**
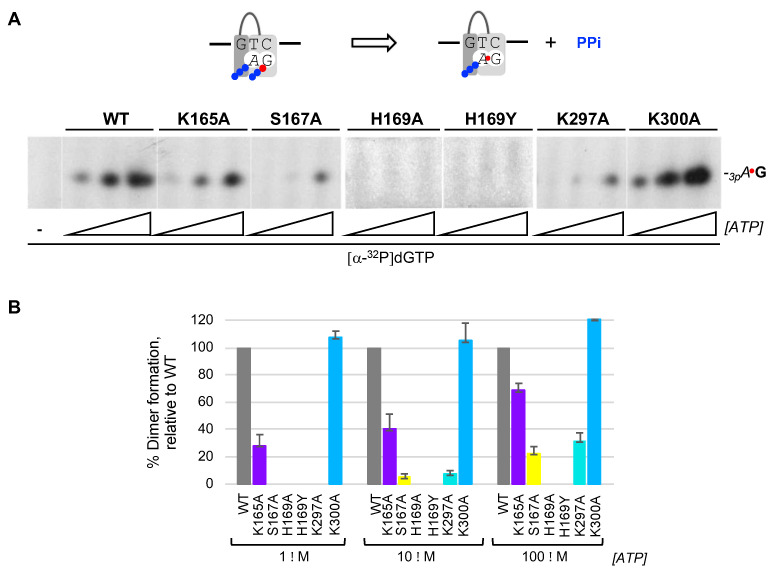
His^169^ is essential for the initial step of PrimPol priming, which is further contributed by other 3′ dNTP ligands. (**A**) Primase assay of dimer formation was performed as indicated in Materials and Methods, by using as template the ssDNA oligonucleotide 3′ T_20_-GTCC-T_36_ 5′ (1 μM), which contains a preferred priming sequence (GTC), WT or mutant *Hs*PrimPols (100 nM), [α-^32^P]dGTP (16 nM; β and γ -phosphates are indicated with a blue ball and α^32^P with a red ball) as the 3′ site nucleotide, and *ATP* (1, 10, 100 μM) as the 5′ nucleotide. (**B**) Quantification of dimer synthesis formation (%) by mutants, relative to WT *Hs*PrimPol (100%), averaged from three different experiments.

**Figure 4 ijms-25-00051-f004:**
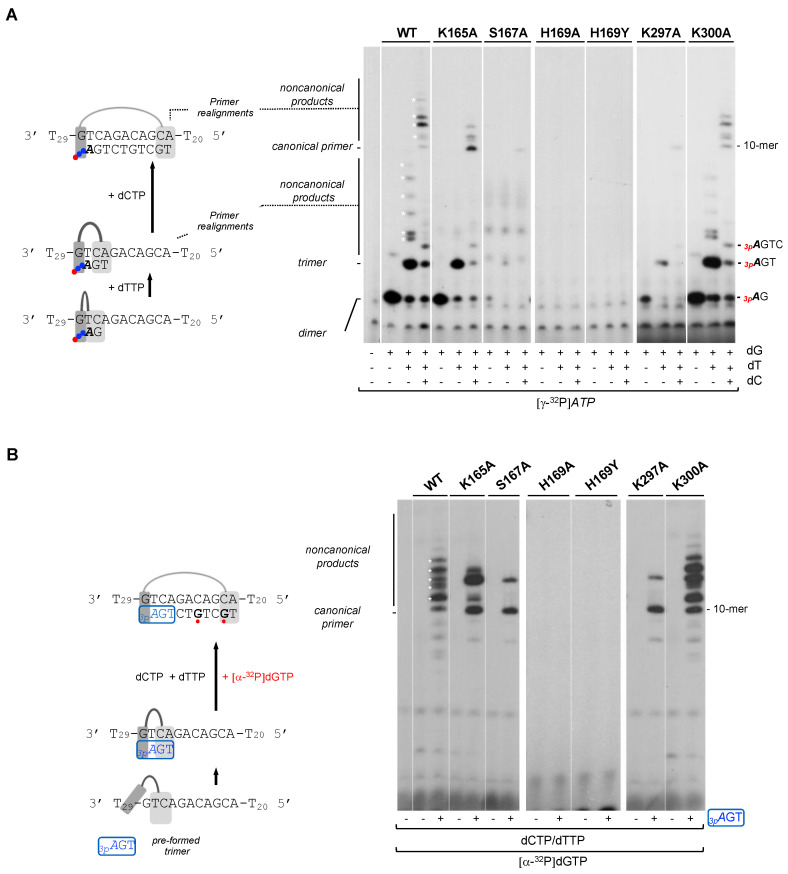
His^169^ is essential for dimer synthesis and primer extension, Ser^167^ and Lys^297^ are required to make dimers but not for primer extension, and Lys^165^, Ser^167^ and Lys^297^ are needed to form non-canonical extension products. (**A**) Dimer formation and further elongation by WT *Hs*PrimPol or the indicated mutants (100 nM) using a template with the preferred priming site (3′ GTC) followed by a heterologous sequence, by sequentially adding *[γ-^32^P]ATP* (16 nM) (*γ-^32^P* is indicated as a red ball) as the 5′ nucleotide, and dGTP, dTTP and dCTP (100 μM) as 3′ incoming dNTPs, in the presence of 1 mM MnCl_2_. (**B**) Elongation of a synthetic mini-primer (*_3p_A*GT in blue with a blue box; 10 μM;) by WT *Hs*PrimPol or the indicated mutants (100 nM) in the presence of dCTP, dTTP (10 μM), [α-^32^P]dGTP (16 nM; in red), and 1 mM MnCl_2_. Labeled dGMP insertion is indicated with a red ball. Non-canonical products are marked by asterisks.

**Figure 5 ijms-25-00051-f005:**
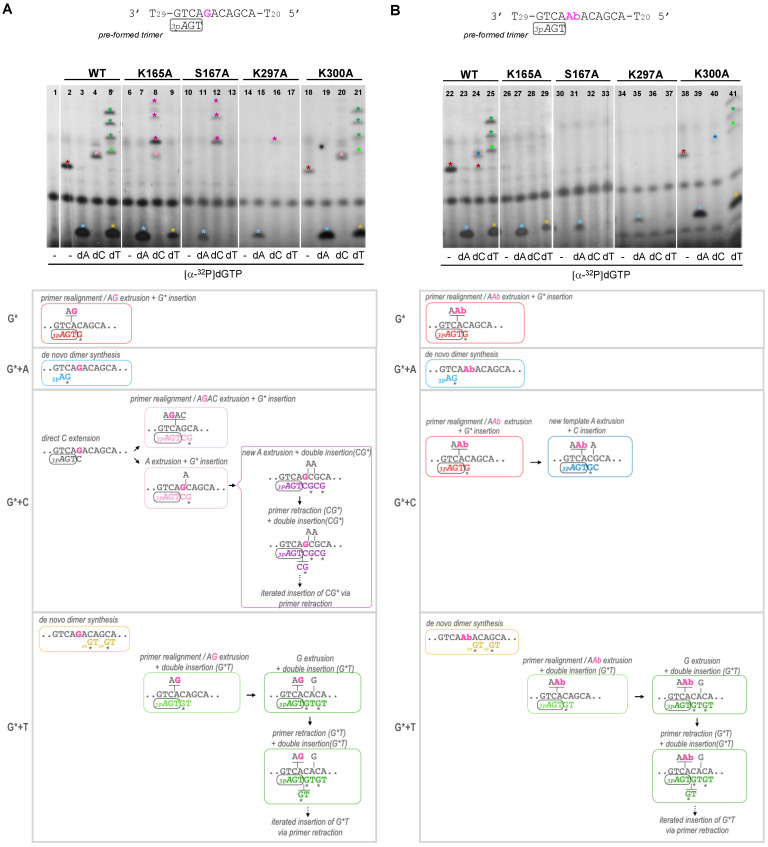
Lys^165^, Ser^167^ and Lys^297^ are necessary to skip an abasic site. Synthetic primer (*_3p_A*GT) extension of WT *Hs*PrimPol and mutants (100 nM) in the presence of either an undamaged template (part **A**) or a template containing an abasic (Ab) site (part **B**), using [α-^32^P]dGTP supplemented with dATP, dCTP or dTTP (100 μM), as indicated, and in the presence of 1 mM MnCl_2_. The upper autoradiograms show the position of different labeled products (indicated by colored asterisks), whereas the lower parts show schemes of their generation (by coloring the primer extension products congruently with the asterisks mentioned above).

**Figure 6 ijms-25-00051-f006:**
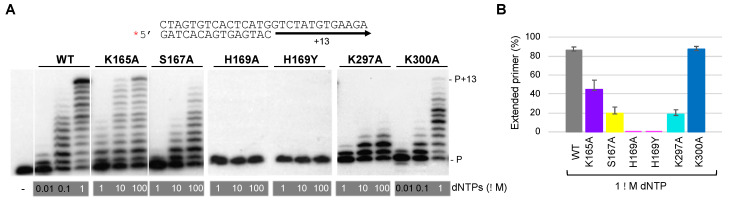
His^169^ is indispensable for PrimPol polymerase activity, whereas other 3′ dNTP ligands have a secondary role. (**A**) Standing-start primer extension assay (standard DNA polymerization on a template/primer, labeled and indicated with a red asterisk) by WT *Hs*PrimPol and 3′ incoming dNTP interactor mutants (50 nM), when using the indicated increasing concentrations of dNTPs, in the presence of 1 mM MnCl_2_. (**B**) Quantification of primer extension (%) of WT, K165A, S167A, K297A and K300A mutants when providing 1 μM dNTPs (average of three different experiments).

**Figure 7 ijms-25-00051-f007:**
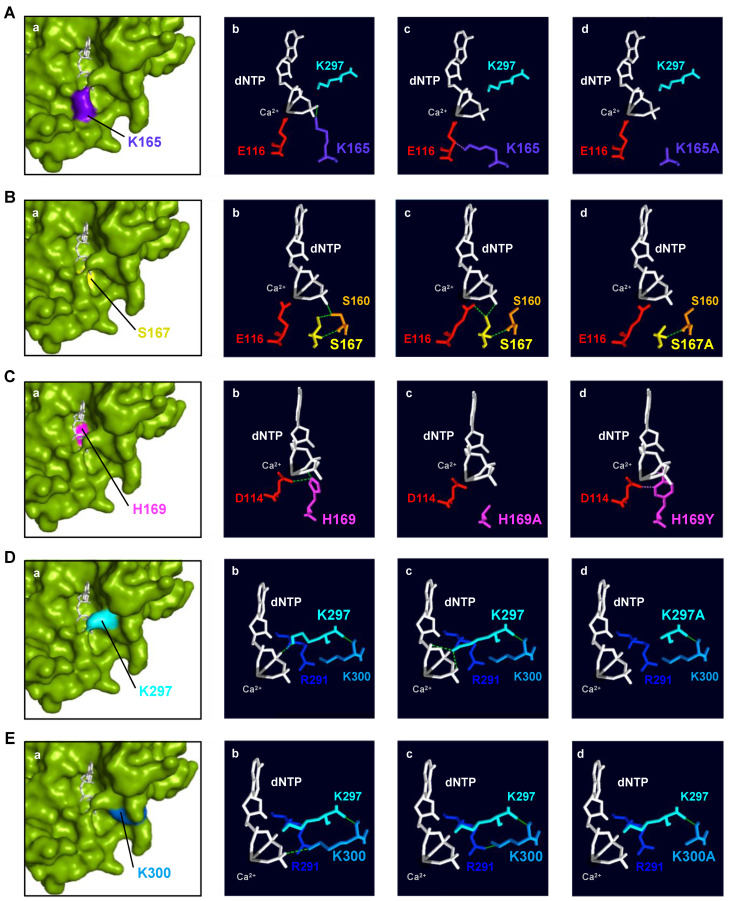
Structural analysis of human PrimPol 3′ incoming dNTP ligands. (**A**) (**a**): Spatial location of Lys^165^ (indigo); (**b**,**c**): interactions with 3′ incoming dNTP (white) and catalytic Glu^116^ (red); (**d**): mutant K165A (modeled). (**B**) (**a**): Spatial location of Ser^167^ (yellow); (**b**,**c**): interactions with Ser^160^ (orange), Glu^116^ (red), and 3′ site incoming dNTP (white); (**d**): mutant S167A (modeled). (**C**) (**a**): Spatial location of His^169^ (magenta); (**b**): interactions with catalytic Asp^114^ (red) and 3′ site incoming dNTP (white); modeled mutants H169A (**c**) and H169Y (**d**). (**D**) (**a**): Spatial location of Lys^297^ (cyan); (**b**,**c**): interactions with 3′ incoming dNTP (white) and Lys^300^ (sky blue); (**d**): mutant K297A (modeled). (**E**) (**a**): Spatial location of Lys^300^ (sky blue); (**b**): interactions with the 3′ incoming dNTP (white); (**c**): interactions with Lys^297^ (cyan) and Arg^291^ (dark blue); (**d**): mutant K300A (modeled). In all panels, green lines indicate hydrogen bonds while pink ones show steric clashes. Ca^2+^ as metal B in the catalytic center is shown as a sphere in grey. PDBid: 5L2X.

**Table 1 ijms-25-00051-t001:** Kinetic parameters for dNTP incorporation opposite the complementary template base (T, G, C, A) of WT PrimPol versus K165A, S167A, K297A and K300A mutants.

Template	dNTP	Enzyme	Km (μM)	Affinity Ratio	Kcat (s^−1^)	Kcat Ratio	Kcat/Km (s^−1^·μM^−1^)	Relative Activity (%)
T	dATP	WT	0.728 ± 0.167	1	0.110 ± 0.019	1	0.1514	100
		K165A	3.775 ± 0.118	0.193	0.079 ± 0.011	0.71	0.0208	13.76
		S167A	51.285 ± 8.165	0.014	0.036 ± 0.005	0.32	0.0007	0.46
		K297A	44.900 ± 8.595	0.016	0.012 ± 0.002	0.11	0.0003	0.17
		K300A	1.053 ± 0.081	0.692	0.077 ± 0.001	0.70	0.0733	48.42
G	dCTP	WT	0.300 ± 0.060	1	0.117 ± 0.012	1	0.3913	100
		K165A	5.953 ± 1.020	0.050	0.113 ± 0.006	0.97	0.0190	4.87
		S167A	67.797 ± 4.196	0.004	0.042 ± 0.006	0.36	0.0006	0.16
		K297A	23.142 ± 4.867	0.013	0.020 ± 0.001	0.17	0.0009	0.22
		K300A	2.050 ± 0.242	0.146	0.094 ± 0.006	0.80	0.0457	11.67
C	dGTP	WT	0.599 ± 0.099	1	0.116 ± 0.015	1	0.1931	100
		K165A	4.955 ± 0.787	0.121	0.057 ± 0.004	0.49	0.0114	5.92
		S167A	50.017 ± 5.430	0.012	0.035 ± 0.009	0.30	0.0007	0.36
		K297A	25.303 ± 3.790	0.024	0.013 ± 0.002	0.11	0.0005	0.27
		K300A	0.530 ± 0.119	1.130	0.055 ± 0.005	0.47	0.1033	53.52
A	dTTP	WT	0.259 ± 0.052	1	0.083 ± 0.001	1	0.3215	100
		K165A	4.324 ± 0.515	0.060	0.060 ± 0.004	0.72	0.0138	4.30
		S167A	167.004 ± 17.283	0.002	0.027 ± 0.005	0.32	0.0002	0.05
		K297A	91.442 ± 17.290	0.003	0.016 ± 0.002	0.20	0.0002	0.06
		K300A	0.481 ± 0.087	0.539	0.098 ± 0.004	1.18	0.2042	63.53

The affinity ratio for the incoming dNTPs (affinity (mutant)/affinity(WT)) is calculated as the ratio Km(WT)/Km (mutant). The kcat ratio is calculated as the ratio kcat (mutant)/kcat(WT). Relative activity (%) is calculated as kcat/Km(mutant)*100/kcat/Km(WT).

## Data Availability

COBALT (Constraint-based Multiple Alignment Tool) is an open-source program available from NBCI (https://www.ncbi.nlm.nih.gov/tools/cobalt/cobalt.cgi). The Swiss-PdbViewer (DeepView) program is also an open-source program (https://spdbv.unil.ch/).

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
