# Peer review of "3′dNTP Binding Is Modulated during Primer Synthesis and Translesion by Human PrimPol"

_ijms, 2023, doi:10.3390/ijms25010051_

Round 1
Reviewer 1 Report
Comments and Suggestions for Authors
The manuscript by Velázquez-Ruiz and coworkers is a well-executed and thoughtful structure-function study of the role of conserved amino acids in the catalytic cycle of human PrimPol. In the opinion of this reviewer, this study is relevant to a general audience and researchers in the field. The following comments are aimed to clarify some points.
1.- Regarding the phrase “PrimPol is a non-conventional primase since it synthesizes preferably DNA primers over RNA primers [2]”. Although PrimPol has a clear preference to incorporate dNTPs, the author should state that the 5´nucleotide is invariably a ribonucleotide (Introduction)
2.- The phrase “In this work we have explored why HsPrimPol has so many 3’ incoming dNTP putative ligands, and if some of them have specific functions during DNA primer synthesis” may be rewritten, a possibility is:
“In this work we have explored why HsPrimPol has a large number of potential 3’ incoming dNTP interacting residues, and if some of them have specific functions during DNA primer synthesis”.
3.- Regarding the sentence “that proposed ModN in the N-terminal of Prim-Pol as responsible for the main template interactions.”. Please explain the meaning of ModN with the basis on the original manuscript “Structure and mechanism of human PrimPol, a DNA polymerase with primase activity” as the Module N-terminal of the catalytic Core.
Figure 1A can be used to delineate the boundaries of ModN and ModC, as well.
4.- Figure 2D shows the ability of HsPrimPol to bind to a radioactively labeled dNTP by itself (binary complex) and also shows the formation of a pre-ternary complex HsPrimPol-ssDNA-dNTP. Please label each of the complexes in the native gel. A band between both complexes is observed in wild-type HsPrimPol, please indicate the identity of this third complex.
5.- The data in figure 3 nicely shows that residues His169, S167 and K297 are necessary for dimer synthesis, whereas K300 is not involved in this step. This data is clear and well-presented.
6.- The sentence “To further evaluate their residual capacity to make DNA primers, especially at a more physiological concentration (100 μM) of 3’ incoming dNTPs, [γ-32P]ATP (16 nM) was used as 5’ site labeled nucleotide” is counterintuitive as ribonucleotides are present in larger concentrations than deoxynucleotides. Please clarify.
7.- The data in Figure 4 support the role of residues Ser167 and Lys297 to stabilize the dGTP during dimer synthesis but not during its elongation. Meaning that using a synthetic mini-primer (3pAGT) mutations in those residues show catalytic activity. In contrast residue His169 is essential at both steps of dimer synthesis and its elongation. In this sense, human PrimPol is predicted to alter its conformation from dimer synthesis to trimer or tetramer (longer nucleotides) synthesis. The authors should speculate how a residue is involved in the binding of the first dNTP (dimer formation) but not in the binding of the next dNTPs. It is possible that the role of these residues (Ser167 and Lys297) is masked because the trimer and tetramer present new interactions with PrimPol.
8.- The data depicted in Figures 5A and 5B (and nicely illustrated in the diagram below) illustrates that residue Lys165 provides extra stabilization to bypass 8oxo Guanosine and abasic sites. The data is precise.
9.- The authors execute a “tour de force” analysis of several mutants measuring the catalytic parameters of human PrimPol. This segment of the manuscript is impressive and the role of the studied residue in assembling a catalytic competent active site and dNTP binding is demonstrated.
Reviewer 2 Report
Comments and Suggestions for Authors
PrimPol is a multi-functional DNA polymerase thought to synthesize DNA ahead of lesions to minimize replication fork collapse. A ternary structure of PrimPol with a primer-template and incoming DNA bound provides the basis for testing amino acids that may be involved in each of these steps. Furthermore, to my knowledge structures of lesion bypass have not been solved so structure-function biochemistry is needed to better understand active site mechanisms. This manuscript completed a detailed structure-function analysis of PrimPol active site amino acids involved in dNTP binding and catalysis during initiation, elongation and lesion bypass.
In general, the study is logical and detailed in analysis of the role of a core set of active site amino acids in incorporation. The writing is clear for the most part. The results and conclusions will contribute to the field's understanding of PrimPol. I recommend publication after considering some minor suggestions below:
Line 140 "migrates slower"
For clarity please include "migrates slower on SDS-PAGE"
Line 170 Typo: "stablishing" should be establishing
Lane 182
You make the point that some primases can bind a dNTP prior to binding ssDNA. Please comment if this is biologically relevant and if in the cell PrimPol is likely pre-bound with a dNTP or not.
Figure 4.
The cartoons on the side are simple and effective. However the asterisks marking the gel bands are not clear. Because there are multiple bands representing different parts of the reaction, the notations need to be more clear.
Figure 5A.
In lane 34, why does WT apparently stall with dG, dT, and dC? Why doesn't incorporation proceed until the end?
Figure 5B.
This is a well designed experiment however the presentation of all the different products and intermediates is confusing and the color scheme does not effectively relate the intermediate to the band on the gel. I would suggest making it more schematic and simple if possible. A cartoon and then sequence details would be more intuitive.
Line 382. Please be more specific in your prediction of "Lys165 provides extra anchorage for the dNTP..."
Line 440. Typo 0,26% to 0.26%
Section 2.7
The structural analysis makes since but you are using a model to interpret the biochemical results. In the text, make sure to make clear that these rotamers are models not actual structures.
Figure 7.
I prefer a white background for panels however, I'll leave it to the authors to decide.
If metals are in the structure with the incoming dNTP, please put them in the figure.
Are there statistical measures to assess the rotamers presented in the panels as the most likely conformation to detail how you chose those conformations.
